# Micro-Osteoperforations Accelerate Tooth Movement without Exacerbating the Progression of Root Resorption in Rats

**DOI:** 10.3390/biom14030300

**Published:** 2024-03-02

**Authors:** Tadasu Sugimori, Masaru Yamaguchi, Jun Kikuta, Mami Shimizu, Shinichi Negishi

**Affiliations:** Department of Orthodontics, School of Dentistry at Matsudo, Nihon University, 2-870-1 Sakaecho-Nishi, Matsudo 271-8587, Japan; sugimori.tadasu@nihon-u.ac.jp (T.S.); kikuta.jun@nihon-u.ac.jp (J.K.); negishi.shinichi@nihon-u.ac.jp (S.N.)

**Keywords:** micro-osteoperforations, orthodontic tooth movement, root resorption, apoptosis

## Abstract

A recent study reported that micro-osteoperforations (MOPs) accelerated tooth movement by activating alveolar bone remodeling. However, very little is known about the relationship between MOPs and external apical root resorption during orthodontic treatment. In this study, in order to investigate the mechanism through which MOPs accelerate tooth movement without exacerbating the progression of root resorption, we measured the volume of the resorbed root, and performed the terminal deoxynucleotidyl transferase (TdT)-mediated dUTP-biotin nick-end labeling (TUNEL) method on exposed MOPs during experimental tooth movements in rats. Male Wistar rats (11 weeks old) were divided into three groups: 10 g orthodontic force (optimal force) applied to the maxillary first molar (optimal force: OF group), 50 g orthodontic force application (heavy force: HF group), and 10 g force application plus three small perforations of the cortical plate (OF + MOPs group). On days 1, 4, 7, 10, and 14 after force application, the tooth movement and root volume were investigated by micro-computed tomography. Furthermore, the number of apoptotic cells in the pressured sides of the periodontal ligament (PDL) and surrounding hard tissues were determined by TUNEL staining. The OF + MOPs group exhibited a 1.8-fold increase in tooth movement on days 7, 10, and 14 compared with the OF group. On days 14, the HF group had a higher volume of root loss than the OF and OF + MOPs groups. On the same day, the number of TUNEL-positive cells in the HF group increased at the root (cementum) site whereas that in the OF group increased at the alveolar bone site. Furthermore, the number of TUNEL-positive cells in the OF + MOPs group increased at the alveolar bone site compared with the OF group. These results suggest that MOPs accelerate orthodontic tooth movement without exacerbating the progression of root resorption.

## 1. Introduction

A long duration of orthodontic treatment may add to the risk of adverse effects such as spot lesions and dental caries, gingivitis, periodontitis, and external apical root resorption [1,2,3]. Moreover, an extended treatment duration could potentially diminish patient adherence and negatively impact oral health-related quality of life, particularly in adults [4]. Furthermore, in the present day, there is a growing cohort of adult patients seeking orthodontic interventions primarily for aesthetic reasons, with a short treatment duration being a key request driven by aesthetic and social considerations [5]. Therefore, shorter treatment durations may provide benefits to both treatment providers and patients.

Accelerating orthodontic tooth movement (OTM) and reducing treatment duration might not only alleviate discomfort but also mitigate potential dental and periodontal complications in patients [6,7]. Regarding this matter, several non-surgical methods (such as photobiomodulation [8,9,10]) and surgical methods (such as corticotomy [11], piezocision [12,13], and periodontally accelerated osteogenic orthodontics [14,15]) have been suggested to hasten orthodontic tooth movement (OTM) with the aim of shortening the total duration of orthodontic treatments. These methods are effective for tooth movement, but their major drawback is their high invasiveness. Recently, a method of accelerating tooth movement called micro-osteoperforations (MOPs) was introduced to activate alveolar bone remodeling while creating minimal surgical trauma. MOPs are a less intrusive alternative to other surgical procedures. A study conducted by Alikahni et al. [16] demonstrated that MOPs accelerated tooth movement by approximately 2.3 times compared to traditional orthodontic methods. Sugimori et al. [17] reported that the activation of cell proliferation and apoptosis of the periodontal ligament (PDL) are involved in the acceleration of tooth movement.

Orthodontic root resorption (ORR) is an inevitable outcome of OTM. Various factors associated with orthodontic treatment and ORR have been identified: extended treatment times [18], heavy force [19], and patient-related risk factors such as genetic predisposition [20], age [21], root irregularities [19], previous tooth injuries [22], and allergies [23,24]. From a cytological perspective, Minato et al. [25] reported that the apoptosis of cementoblasts is involved in the onset and exacerbation of root resorption. In addition, it is believed that excessive tooth movement induces root resorption. Accelerating tooth movement due to a strong load also causes root resorption [26]. Chandorikar and Bhad [3] reported that the overall root resorption after treating with MOPs was higher in the treated than the control group. Contrarily, a study by Dos Santos et al. conducted in 2020 [27] demonstrated that MOPs did not affect root resorption. Furthermore, sporadic clinical observations have indicated that the use of MOPs does not worsen the progression of root resorption [28]. Whether increasing tooth movement velocity with surgery procedures causes root resorption is not clear.

This study focused on the apoptosis of PDL cells and cementoblasts, which is believed to be related to the acceleration of tooth movement by MOPs and the progression of root resorption. This study aimed to investigate the mechanism by which MOPs accelerate tooth movement without exacerbating the progression of root resorption by measuring the volume of the resorbed root and performing the terminal deoxynucleotidyl transferase (TdT)-mediated dUTP-biotin nick-end labeling (TUNEL) method on exposed MOPs during experimental tooth movements in rats.

## 2. Materials and Methods

### 2.1. Animals

The animal experimental protocol used in this study was approved by the Ethics Committee for Animal Experiments at the Nihon University School of Dentistry at Matsudo (approval No. AP20MAS011-2). A total of fifteen male 11-week-old Wistar rats (Sankyo Labo Service, Inc., Tokyo, Japan; body weight: 300 ± 30 g) were used for the experiments. The animals were maintained in the animal center of Nihon University School of Dentistry at Matsudo in separate cages in an environment with a 12 h light/dark cycle and a constant temperature of 23 °C and were provided with food and water ad libitum. During the experimental period, powered food was provided for all the animals. The health status of each rat was evaluated by daily body weight monitoring for 1 week before the start of the experiments. The rats were randomly divided into 3 groups according to the treatment given: 10 g orthodontic force (optimal force: OF) applied to the maxillary first molar (OF group), 10 g force application plus 3 small perforations of the cortical plate (OF+MOP group), and 50 g orthodontic force (heavy force: HF) applied to the maxillary first molar (HF group).

### 2.2. Application of Orthodontic Devices

In each group, pentobarbital sodium (40 mg/kg body weight) was used to anesthetize the animals prior to the application of the orthodontic devices. Following the method outlined by Nakano et al. [29], we employed a closed-coil spring (wire size: 0.005 in; diameter: 0.083 in; Accurate, Inc., Tokyo, Japan) secured to the maxillary first molar using a 0.008 in stainless-steel ligature wire (Tomy International, Inc., Tokyo, Japan). The opposite end of the coil spring was similarly secured, with the holes in the maxillary incisors drilled slightly above the gingival papilla using a 1/4 round bur and the same ligature wire. Movement of the upper first molar toward the front was achieved using a closed-coil spring with forces of either 10 or 50 g. The experiment spanned a period of 14 days (Figure 1).

### 2.3. Surgical Procedure

MOPs were induced using the method described by Teixeira et al. [30]. In the OF + MOPs group, the animals received 3 shallow perforations, approximately 0.25 mm in diameter (0.25 mm depth), 5 mm mesial to the first molar using a 1/4 round bur and handpiece (Figure 2).

### 2.4. Tissue Preparation

The experiment lasted for 14 days following the initiation of tooth movement. The animals were deeply anesthetized using pentobarbital sodium and then underwent transcardial perfusion with a solution of 4% paraformaldehyde in 0.1 M phosphate buffer. Subsequently, the maxilla was promptly dissected and immersed in the same fixative for 18 h at 4 °C. To prepare the specimens, they were decalcified in a 10% (*w*/*v*) solution of ethylenediaminetetraacetic acid disodium salt (pH 7.4) for 4 weeks. Afterward, the decalcified specimens underwent dehydration through an ethanol series and were embedded in paraffin using conventional methods. Each sample was sliced into 4 μm sections horizontally and then processed for hematoxylin and eosin (HE) as well as TUNEL staining. The evaluation focused on the periodontal tissues in the mesial part of the distal buccal root of the first upper molar. Animals that did not display tooth movements were included in the control group (Figure 3).

### 2.5. Tooth Movement Measurement

To measure tooth movement accurately, we used micro-CT to assess the distance between the enamel at the furthest point of the first molar and the closest point of the second molar in each animal. An in vivo micro-CT system (Rigaku-µCT^®^, Tokyo, Japan) was employed for quantitative image analysis of the tooth movement. Rat molars underwent scanning with a µ-CT using an X-ray source of 90 kV/88 µA on days 0, 1, 4, 7, 10, and 14. Rats, deeply anesthetized with intraperitoneally administered sodium pentobarbital (35 mg/kg), were positioned on the stage, and the imaging involved a full 360° rotation of the sample, each lasting 17 s. We selected an isotropic resolution of a 30 × 30 × 30 µm voxel size to depict the microstructure of the interdental spaces between the first (M1) and second (M2) molars. The raw 3D images were visualized and analyzed using I-View^®^ software ver. 2.0 (J. Morita, Kyoto, Japan).

### 2.6. Measurement of Root Resorption Volume by Micro-CT

The teeth were subsequently extracted from the images using software, and the bone was removed. The maxillary first molars of rats have five roots. Their distobuccal roots were analyzed. Custom programming of the software, based on a convex hull algorithm similar to that used by Harris [31], was adopted to analyze the volume of the root resorption craters on the root surface.

The distobuccal roots were trimmed (since they are considered to have the smallest individual differences) parallel to the occlusal surface of the molar, 0.5 mm from the root bifurcation towards the root apex, and the volume of the tooth root was measured using the bone structure analysis software TRI/3D-BON (Ratoc System Engineering, Tokyo, Japan). The difference between the volumes on days 14 and 0 was calculated as the amount of root reduction.

### 2.7. TUNEL Staining

TUNEL staining was performed using an Apoptosis Detection Kit (Trevigen, Gaithersburg, MD, USA). The 4 μm paraffin sections underwent an incubation step with proteinase K (Trevigen), diluted at a ratio of 1:200, at 37 °C for 15 min. Following this, they were rinsed with deionized water, treated with 3% H_2_O_2_ for 5 min, and subsequently washed again with deionized water. For the positive control, the sections were exposed to DNase I at 37 °C for 30 min. All sections were then subjected to incubation with terminal deoxynucleotide transferase (TdT) and biotin-dUTP at 37 °C for 1 h, except for the negative control sections where the TdT enzyme was omitted. After a washing step, the sections were incubated in a Strep–HRP solution at 37 °C for 10 min. Following another wash, the apoptotic cells were marked using diaminobenzidine (DAB). These sections were counterstained with 1% methyl green, dehydrated, and finally mounted.

### 2.8. Statistical Methods

The values are expressed as mean ± standard deviation (S.D.) for each group in each figure. A Mann–Whitney U-test was employed to compare the means of the groups, and *p* < 0.05 was considered to indicate statistical significance.

## 3. Results

### 3.1. Tooth Movement during Experimental Period

There were no excessive body weight changes in any of the animals. No significant difference was observed between the OF and HF groups in tooth movement. The OF + MOPs group exhibited 1.79-, 1.84-, and 1.81-fold increases in tooth movement compared with the OF group on days 7, 10, and 14, respectively (*p* < 0.05) (Figure 4). 

### 3.2. Measurement of Root Resorption Volume by Micro-CT

On day 14, the crater volumes at the root surfaces were compared among the three groups. Furthermore, 3D images of the distobuccal roots were reconstructed. In the OF and OF + MOPs groups, a slightly obvious root resorption was observed. Both wide shallow and deep resorption craters were observed in the HF group (Figure 5A). The total root resorption volumes were significantly greater in the HF group than in the OF and OF + MOPs groups (*p* < 0.05). No significant difference was observed between the OF and OF + MOPs groups in terms of resorption volume (Figure 5B). 

### 3.3. Histological Changes in Periodontal Tissues during Tooth Movement (H&E Staining)

In the OF group, the periodontal ligament (PDL) specimens were composed of relatively dense connective tissue fibers and fibroblasts that regularly ran in a horizontal direction from the root cementum towards the alveolar bone. On the alveolar bone surface, bone resorption lacunae with multinucleate osteoclasts were founded. In contrast, root resorption lacunae with multinucleated odontoclasts were not observed on the root surface (Figure 6). 

In the OF + MOPs group, the arrangement of the fibers and fibroblasts became coarse and irregular and the blood capillaries were compressed on day 14. Resorption lacunae with many multinucleate osteoclasts were detected on the alveolar bone surface. In addition, a few odontoclasts on the root were observed (Figure 6).

In the HF group, the PDL was composed of a coarse arrangement of fibers and expanded blood capillaries. Many resorption lacunae with multinucleate osteoclasts were observed on the alveolar bone, and many root resorption lacunae with multinucleated odontoclasts were recognized on the surface of the root (Figure 6).

### 3.4. Ratios of TUNEL-Positive Cells 

In the TUNEL staining on day 14, apoptosis-positive cells were observed at the alveolar bone site in the OF and OF + MOPs groups, and the number of these cells in the latter group was higher compared with the former group. In the HF group, however, apoptosis-positive cells were observed at the root (cementum) site (Figure 7A).

At the alveolar bone site, the number of apoptosis-positive cells in the three groups increased in a time-dependent manner from day 1 to 14. From day 1 to 14, the number of these cells was significantly higher in the OF + MOPs group than in the OF and HF groups (* *p* < 0.05) (Figure 7B). 

At the root site, the number of apoptosis-positive cells was significantly higher in the HF group than in the OF and OF + MOPs groups from day 1 to 14 (* *p* < 0.05). No significant difference was observed between the OF and OF + MOPs groups in terms of the number of apoptosis-positive cells (Figure 7C).

## 4. Discussion

Figure 4 demonstrates that the OF + MOPs group exhibited a 1.8-fold increase in tooth movement compared with the OF group from day 7 to 14 (Figure 4). Cheung et al. [32] showed how MOPs accelerate tooth movement in rats while simultaneously reducing both bone volume and density. They found that the maxillary first molar on the MOP-treated side moved almost twice the distance compared with the untreated side during the treatment period. This increased movement was caused by the demineralization of the bone on the pressured side induced by the MOPs. Bone restructuring, triggered by force applied to the teeth, was investigated by Chang et al. [33], who found that the direction of tooth movement correlated with a more pronounced decrease in alveolar bone density.

It is evident that the decrease in the bone volume to total volume ratio (BV/TV) and bone mineral density (BMD) on the MOP-treated side initiates regulatory processes that facilitate accelerated tooth movement. Our findings are consistent with those of Baloul et al. [34], who reported a significant decrease in BV/TV and BMD after 7 days when tooth movement was combined with alveolar decortication. Consequently, our results further support that the MOPs enhance OTM by stimulating rapid bone remodeling.

Alikhani et al. reported that MOPs significantly increased tooth movement and the levels of inflammatory markers [16]. Furthermore, Frost et al. [35] showcased that the initial injury prompted an acceleration of the typical localized healing mechanisms, termed the regional acceleratory phenomenon (RAP). Typically observed following a fracture, arthrodesis, osteotomy, or bone grafting, RAP potentially entails the mobilization and stimulation of precursor cells essential for wound healing, primarily concentrated at the site of injury. Therefore, inflammation is induced by MOP-activated bone remodeling around the tooth along with orthodontic force. Furthermore, as suggested by recent evidence on this topic, bone tissue and soft tissue healing is a biological process where several factors, such as immune cells, growth factors, and proteins, are involved in repairing and eventually regenerating the damaged tissue [36]. These reports supported the results in this study. 

To measure the volume of root resorption, 3D images of the distobuccal roots were also reconstructed (Figure 5A). There was no significant difference between the OF and OF + MOPs groups in terms of resorption volume. On day 14, the HF group had a significantly greater total root resorption volume than the OF and OF + MOPs groups (Figure 5B). Ru et al. [37] measured the root resorption volume during orthodontic movement in three dimensions using in vivo micro-CT and reported that the resorption volume of the mesial root significantly increased on day 7 of orthodontic loading (10 g). Furthermore, Gonzales et al. [38] demonstrated that force magnitude and duration affect tooth movement and root resorption in rat molars. The largest and deepest resorption craters were found in the distobuccal root in the HF group. Interestingly, although the MOPs group had the greatest movement distance, it had a lower root resorption volume than the HF group (Figure 5). Also, Figure 6 shows that resorption lacunae were observed on the root side of the PDL in the HF group, whereas in the OF and OF + MOPs groups, they were observed on the alveolar bone side of the PDL (Figure 6). Recently, Tsai et al. [39] revealed that MOP-facilitated tooth movement acceleration resulted in decreased root resorption according to the HE analysis. Similarly, Cheung et al. [32] reported that a volumetric analysis of all five roots of the maxillary first molar during rat experimental tooth movement revealed no significant increase in root resorption with MOPs. These results indicated that MOPs accelerate tooth movement without exerting adverse effects on the roots. 

Cell death was initially thought to be the result of one of two distinct processes: apoptosis (also known as programmed cell death) or necrosis (uncontrolled cell death). Apoptosis is characterized by several characteristic morphological changes in the cell structure, together with some enzyme-dependent biochemical processes. Thus, the mechanism by which MOPs prevent root resorption was explored via apoptosis staining. Apoptosis staining using the TUNEL assay has been employed to detect DNA breakage in studies unrelated to cell death [40,41].

In the TUNEL staining of this study, apoptosis-positive cells were observed at the alveolar bone site of the OF and OF + MOPs groups on day 14, and the number of these cells increased in the former group compared with the latter group. In the HF group, however, apoptosis-positive cells were observed at the root (cementum) site. The number of apoptosis-positive cells in the alveolar bone site was significantly higher in the OF + MOPs group than in the OF and HF groups from day 1 to 14. At the root site, the number of apoptosis-positive cells was significantly higher in the HF group than in the OF and HF groups from day 1 to 14. These results suggest that apoptosis of the PDL, cementum, and osteocytes stimulate tooth root and alveolar bone resorption during tooth movement with MOPs (Figure 7).

OTM has been shown to trigger apoptosis in periodontal tissues soon after force application. Several studies have reported that compressive force application leads to the death of various periodontal cells, including PDL cells [42], osteocytes [43,44], and cementocytes (as observed in the work of Matsuzawa et al. [45]), in experimental tooth movement models.

Regarding the role of apoptosis in the occurrence of root resorption, Minato et al. [25] found a significant increase in the number of caspase 3- and caspase 8-positive cells, as well as receptor activator of nuclear factor-kappa B ligand (RANKL)-positive cells, in the HF (50 g) group compared with the OF (10 g) group. They observed that root resorption occurred after the induction of apoptosis in the cementum due to heavy force application. These observations strongly indicate the involvement of cementoblast apoptosis in root resorption. Furthermore, Wang et al. [46] observed a significant increase in the expression levels of caspase 3 specifically within the PDL on the alveolar bone side following OTM using a light force (0.392 N) in their rat experimental model. In a similar model, Rana et al. [47] observed TUNEL-positive staining in compressed PDLs, indicating cellular apoptosis. In addition, Sugimori et al. [17] reported that the number of TUNEL-positive cells in the OF (10 g) + MOPs group increased on days 1 and 7 compared with the OF (50 g) group. Consequently, apoptosis was identified as a significant contributor to OTM, potentially enhancing the pace of tooth movement and the root resorption process.

Regarding the association between OTM acceleration and root resorption, Shahrin et al. [28] reported that OTM acceleration with MOPs during the alignment phase does not exacerbate root resorption in patients with moderate crowding of the upper labial segment. Alqadasi et al. [48] also reported that MOPs accelerated orthodontic canine retraction and that this technique did not induce root resorption. Aksakalli et al. [49] demonstrated that in a case report using a mini-screw, MOPs accelerated canine retraction without root resorption. Those reports support the results of this study. However, there are also conflicting reports indicating an exacerbation of root resorption. Chan et al. [26] reported that MOPs resulted in greater root resorption on day 28 in a rat experimental model. Furthermore, Al-Attar et al. [50] demonstrated that MOPs exacerbated the progression of root resorption under the same conditions. These reports are data from day 28, and it is necessary to observe this experiment until day 28.

When the optimal force was applied, apoptosis occurred in the PDL on the alveolar bone side, including bone resorption and tooth movement. Also, MOPs resulted in even more apoptosis in the PDL on the alveolar bone side, accelerating tooth movement without causing root resorption. Conversely, when heavy force was applied, apoptosis occurred in the PDL on the cementum side, leading to resorption of the tooth root. Future studies are warranted to determine whether the site of apoptosis varies depending on the magnitude of the force.

## 5. Conclusions

This study shows that MOPs facilitate OTM during orthodontic treatment without exacerbating root resorption by activating apoptosis in the PDL on the alveolar bone side. The results obtained reveal that MOPs, which are the least invasive and painful surgical approach, do not exacerbate root resorption, an unintended consequence of orthodontic treatment, and shorten the treatment duration. Consequently, this is good news for adults who tend to experience slow tooth movement and prolonged treatment duration; in addition, an increase in the number of adults undergoing orthodontic treatment is expected. MOPs may prevent patient frustration and increased risk of dental caries, periodontitis, and the progression of root resorption due to extended orthodontic treatment through accelerating orthodontic tooth movement and shortening the treatment period.

## Figures and Tables

**Figure 1 biomolecules-14-00300-f001:**
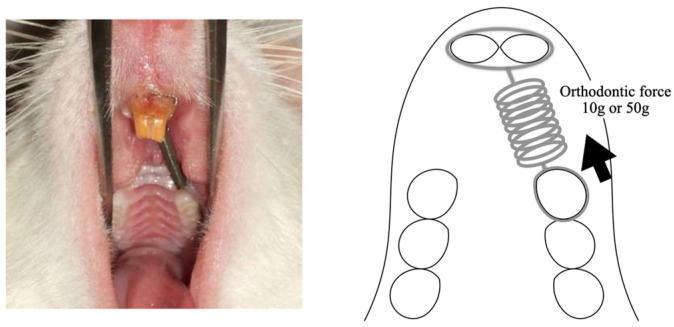
The rat model of orthodontic tooth movement used in the present study. Tooth movement is induced with a closed-coil spring ligated to the maxillary first molar using a 0.008-inch stainless steel ligation wire (wire size, 0.005 inch; diameter, 1/12 inch). The other side of the coil spring is ligated using the same ligation wire to a hole in the maxillary incisor, which is drilled in the cleft just above the gingival papilla using a 1/4 round bur. The maxillary right first molar was moved in the mesial direction via the application of a force of 10 g or 50 g by the sealed coil spring.

**Figure 2 biomolecules-14-00300-f002:**
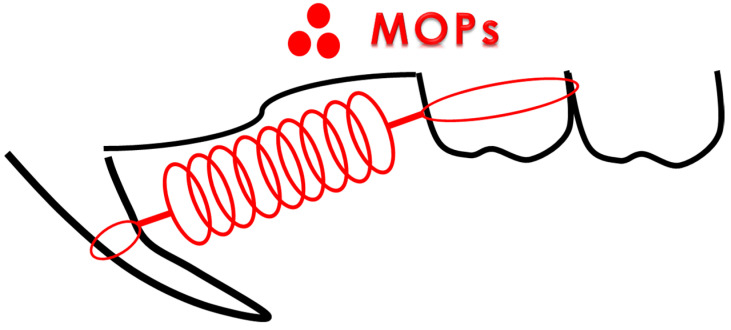
Schematic representation of the location of micro-osteoperforations (MOPs). Rats in the MOP group received 3 shallow perforations, approximately 0.25 mm in diameter (depth of 0.25 mm), on the buccal alveolar bone 5 mm mesial to the maxillary first molar.

**Figure 3 biomolecules-14-00300-f003:**
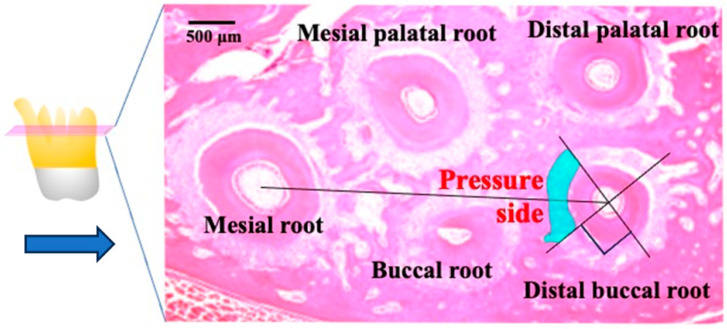
A schematic diagram showing the survey area (shaded box) at the mesial center of the distal root of the maxillary first molar obtained from a rat model of orthodontic tooth movement. The periodontal tissue in the region of compression was defined as the tissue connecting the center of the distobuccal (DB) root and the center of the mesial root (M) of the maxillary first molar, and it comprised one-fourth of the mesial region facing the DB root. The large arrows indicate the direction of force. Root resorption was investigated in 300 µm sections (shaded box) from the area close to the furcation on the mesial surface of the DB root, which was the side of compression during tooth movement. Scale bar = 500 µm.

**Figure 4 biomolecules-14-00300-f004:**
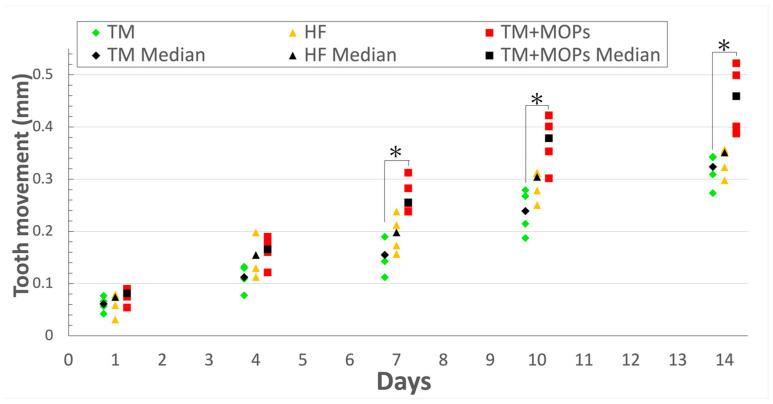
Comparison of the tooth movement distance. The OF + MOPs group exhibited a 1.79-, 1.84-, and 1.81-fold increase in tooth movement compared with the OF group at days 7, 10, and 14, respectively. OF group: optimal force group; OF + MOPs group: optimal force + micro-osteoperforations group; HF group: heavy force group. Data are expressed as means ± standard deviations. Tested by Mann–Whitney U test; * *p* < 0.05.

**Figure 5 biomolecules-14-00300-f005:**
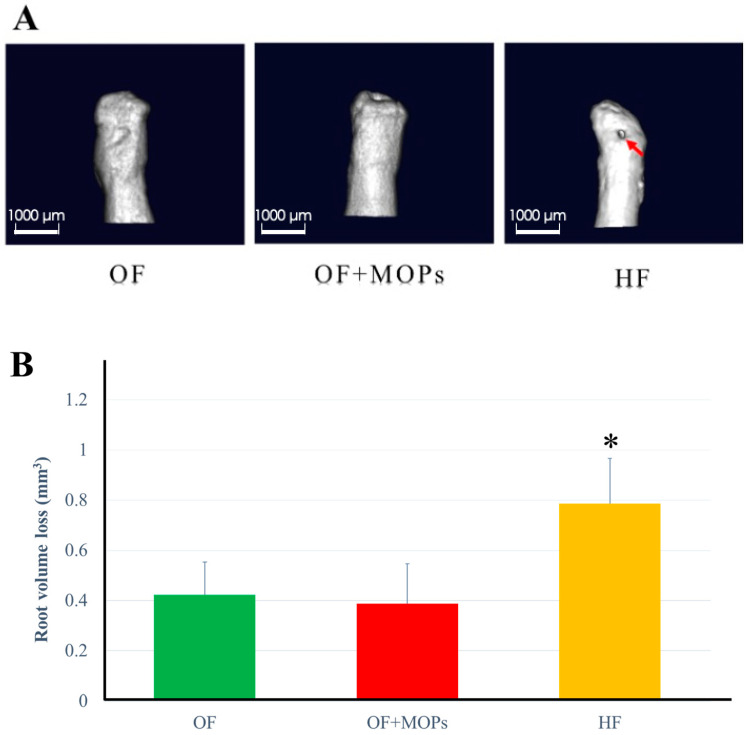
Micro-CT analysis of root resorption. Crater volumes at the distobuccal root surfaces of maxillary first molars were compared among the three groups on day 14. OF group: optimal force group; OF + MOPs group: optimal force + micro-osteoperforations group; HF group: heavy force group. (**A**) In the OF and OF + MOPs groups, there were obvious but low levels of root resorption. Both wide shallow and deep resorption craters were observed in the HF group (red arrow). (**B**) The total root resorption volume in the HF group was significantly greater than that of the OF and OF + MOPs groups. No significant difference was found between the OF and OF + MOPs groups in resorption volume. Data are expressed as means ± standard deviations. Tested by Mann–Whitney U test; * *p* < 0.05.

**Figure 6 biomolecules-14-00300-f006:**
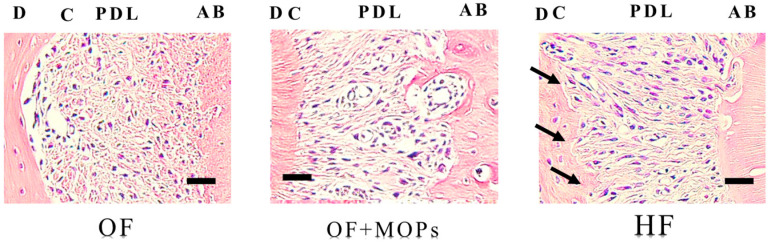
Light microscopic images showing the effects of various orthodontic tooth movement conditions (OF, OF + MOPs, HF) on the expression of multinuclear osteoclasts (hematoxylin and eosin staining, ×400) at 14 days after force application in a rat model of orthodontic tooth movement. The number of odontoclasts (arrows) on the root surfaces in the HF (50 g) group was greater than that on the root surfaces in the OF (10 g) and OF (10 g) + MOPs groups on day 14. OF group: optimal force group; OF + MOPs group: optimal force + micro-osteoperforations group; HF group: heavy force group. The arrows indicate the sites of root resorption. Scale bar = 50 µm. PDL, periodontal ligament; C, cementum; D, dentin; AB, alveolar bone.

**Figure 7 biomolecules-14-00300-f007:**
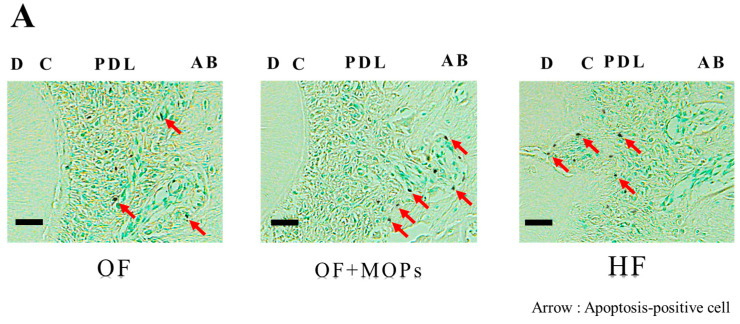
Effects of different orthodontic tooth movement conditions (OF, OF + MOPs, HF) on the number of TUNEL-positive cells at days 1, 3, 5, and 7 after force application in a rat model of orthodontic tooth movement. OF group: optimal force group; OF + MOPs group: optimal force + micro-osteoperforations group; HF group: heavy force group. (**A**) The TUNEL-positive cells at 14 days after force application (OF, OF + MOPs, HF) in a rat model of orthodontic tooth movement. The apoptosis-positive cells of the OF and OF + MOPs groups were observed at the alveolar bone site, and the number of apoptosis-positive cells was higher in the OF+MOP group compared with the OF group. On the other hand, the apoptosis-positive cells of the HF group were observed at the root (cementum) site. The red arrows indicate the osteoclast/odontoclast. PDL, periodontal ligament; C, cementum; D, dentin; AB, alveolar bone. Magnification = ×400. Scale bar = 50 µm. (**B**) The quantitative evaluation of TUNEL-positive cells at 14 days after force application (OF, OF + MOPs, HF) in alveolar bone site in a rat model of orthodontic tooth movement. The number of apoptosis-positive cells in the three groups increased in a time-dependent manner from day 1 to 14. Furthermore, the number of apoptosis-positive cells in the OF + MOPs group was significantly higher than that in the OF and HF groups from day 1 to 14 (* *p* < 0.05). (**C**) The quantitative evaluation of TUNEL-positive cells at 14 days after force application (OF, OF + MOPs, HF) in root site in a rat model of orthodontic tooth movement. The number of apoptosis-positive cells in the HF group was significantly higher than that in the OF and HF group from day 1 to 14. No significant difference was found between the OF and OF + MOPs groups in terms of the number of apoptosis-positive cells. * *p* < 0.05, when comparing to the OF and HF groups.

## Data Availability

Data are available from the author on request.

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
