# Peer review of "Micro-Osteoperforations Accelerate Tooth Movement without Exacerbating the Progression of Root Resorption in Rats"

_biomolecules, 2024, doi:10.3390/biom14030300_

Round 1

Reviewer 1 Report

Comments and Suggestions for Authors

The aim of this manuscript is to investigate the mechanism by which MOPs accelerate tooth movement without exacerbating the progression of root resorption by measuring the volume of resorbed root and performed the immunostaining for the terminal deoxynucleotidyl transferase (TdT)-mediated dUTP-biotin nick-end-labelling method (TUNEL)-exposed MOPs during rat experimental tooth movement.

This manuscript shows rich content, providing a deep insight for some works: the study is within the journal’s scope, and I found it to be well-written, providing sufficient information. Even if the manuscript provides an organic overview, with a densely organized structure and based on well-synthetized evidence, there are some suggestions necessary to make the article complete and fully readable. For these reasons, the manuscript requires major changes.

Please find below an enumerated list of comments on my review of the manuscript:

INTRODUCTION:

LINE 32: Long duration of orthodontic treatment may add the risk of adverse effects of pain, discomfort, and orthodontically induced inflammatory root resorption (OIIRR) (see, for reference: Chandorikar, H., & Bhad, W. A. (2023). Impact of micro-osteoperforations on root resorption and alveolar bone in en-masse retraction in young adults: A CBCT randomized controlled clinical trial. International Orthodontics21(1), 100714).

MATERIALS AND METHODS:

LINE 121: Please, add in figure legend 3 the scalebar.

DISCUSSION:

LINE 301: Furthermore, as suggested by recent evidence on this topic, bone tissue and soft tissue healing is a biological process where several factors, such as immunity cells, growth factors, and proteins, are involved in repairing and eventually regenerating the damaged tissue (see, for reference: Gerardi, D., Santostasi, N., Torge, D., Rinaldi, F., Bernardi, S., Bianchi, S., ... & Varvara, G. (2023). Regenerative Potential of Platelet-Rich Fibrin in Maxillary Sinus Floor Lift Techniques: A Systematic Review. JOURNAL OF BIOLOGICAL REGULATORS & HOMEOSTATIC AGENTS37(5), 2357-2369). This is the major concern of this manuscript: the manuscript may benefit from providing, in this discussive section, an organic overview of the healing process, defined and influenced by distinct factors.

The main topic is interesting, and certainly of great clinical impact. As regards the originality and strengths of this manuscript, this is a significant contribute to the ongoing research on this topic, as it extends the research field on the mechanism by which MOPs accelerate tooth movement without exacerbating the progression of root resorption by measuring the volume of resorbed root and performed the immunostaining for the terminal deoxynucleotidyl transferase (TdT)-mediated dUTP-biotin nick-end-labelling method (TUNEL)-exposed MOPs during rat experimental tooth movement. Overall, the contents are rich, and the authors also give their deep insight for some works.

As regards the section of methods, there is a specific and detailed explanation for the methods used in this study: this is particularly significant, since the manuscript relies on a multitude of methodological and statistical analysis, to derive its conclusions. The methodology applied is overall correct, the results are reliable and adequately discussed.

The conclusion of this manuscript is perfectly in line with the main purpose of the paper: the authors have designed and conducted the study properly. As regards the conclusions, they are well written and present an adequate balance between the description of previous findings and the results presented by the authors.

Finally, this manuscript also shows a basic structure, properly divided and looks like very informative on this topic. Furthermore, figures and tables are complete, organized in an organic manner and easy to read.

In conclusion, this manuscript is densely presented and well organized, based on well-synthetized evidence. The authors were lucid in their style of writing, making it easy to read and understand the message, portrayed in the manuscript. Besides, the methodology design was appropriately implemented within the study. However, many of the topics are very concisely covered. This manuscript provided a comprehensive analysis of current knowledge in this field. Moreover, this research has futuristic importance and could be potential for future research. However, major concerns of this manuscript are with the introductive, methodological and discussive sections: for these reasons, I have major comments for these sections, for improvement before acceptance for publication. The article is accurate and provides relevant information on the topic and I have some major points to make, that may help to improve the quality of the current manuscript and maximize its scientific impact. I would accept this manuscript if the comments are addressed properly.

Author Response

Manuscript ID: biomolecules-2826766 - Major Revisions

Response to Reviewer 1 Comments

Thank you for giving us valuable advice and comments regarding our manuscript. We revise upon the manuscript as follows.

Reviewer 1

Comments and Suggestions for Authors

The aim of this manuscript is to investigate the mechanism by which MOPs accelerate tooth movement without exacerbating the progression of root resorption by measuring the volume of resorbed root and performed the immunostaining for the terminal deoxynucleotidyl transferase (TdT)-mediated dUTP-biotin nick-end-labelling method (TUNEL)-exposed MOPs during rat experimental tooth movement.

This manuscript shows rich content, providing a deep insight for some works: the study is within the journal’s scope, and I found it to be well-written, providing sufficient information. Even if the manuscript provides an organic overview, with a densely organized structure and based on well-synthetized evidence, there are some suggestions necessary to make the article complete and fully readable. For these reasons, the manuscript requires major changes.

Please find below an enumerated list of comments on my review of the manuscript:

INTRODUCTION:

Point 1:

LINE 32: Long duration of orthodontic treatment may add the risk of adverse effects of pain, discomfort, and orthodontically induced inflammatory root resorption (OIIRR) (see, for reference: Chandorikar, H., & Bhad, W. A. (2023). Impact of micro-osteoperforations on root resorption and alveolar bone in en-masse retraction in young adults: A CBCT randomized controlled clinical trial. International Orthodontics21(1), 100714).

Response 1:

Thank you for the pointing out.

As you pointed out, we have revised “Introduction” as follows,

P1, L31-38.

MATERIALS AND METHODS:

Point 2:

LINE 121: Please, add in figure legend 3 the scalebar.

Response 2:

Thank you for the pointing out.

As you pointed out, we have revised in Figure 3.

DISCUSSION:

Point 3:

LINE 301: Furthermore, as suggested by recent evidence on this topic, bone tissue and soft tissue healing is a biological process where several factors, such as immunity cells, growth factors, and proteins, are involved in repairing and eventually regenerating the damaged tissue (see, for reference: Gerardi, D., Santostasi, N., Torge, D., Rinaldi, F., Bernardi, S., Bianchi, S., ... & Varvara, G. (2023). Regenerative Potential of Platelet-Rich Fibrin in Maxillary Sinus Floor Lift Techniques: A Systematic Review. JOURNAL OF BIOLOGICAL REGULATORS & HOMEOSTATIC AGENTS37(5), 2357-2369). This is the major concern of this manuscript: the manuscript may benefit from providing, in this discussive section, an organic overview of the healing process, defined and influenced by distinct factors.

Response 3:

Thank you for the pointing out.                                                                                                                

As you pointed out, we have added in “Discussion”.

P10, L327-331.

Point 4:

The main topic is interesting, and certainly of great clinical impact. As regards the originality and strengths of this manuscript, this is a significant contribute to the ongoing research on this topic, as it extends the research field on the mechanism by which MOPs accelerate tooth movement without exacerbating the progression of root resorption by measuring the volume of resorbed root and performed the immunostaining for the terminal deoxynucleotidyl transferase (TdT)-mediated dUTP-biotin nick-end-labelling method (TUNEL)-exposed MOPs during rat experimental tooth movement. Overall, the contents are rich, and the authors also give their deep insight for some works.

As regards the section of methods, there is a specific and detailed explanation for the methods used in this study: this is particularly significant, since the manuscript relies on a multitude of methodological and statistical analysis, to derive its conclusions. The methodology applied is overall correct, the results are reliable and adequately discussed.

The conclusion of this manuscript is perfectly in line with the main purpose of the paper: the authors have designed and conducted the study properly. As regards the conclusions, they are well written and present an adequate balance between the description of previous findings and the results presented by the authors.

Finally, this manuscript also shows a basic structure, properly divided and looks like very informative on this topic. Furthermore, figures and tables are complete, organized in an organic manner and easy

In conclusion, this manuscript is densely presented and well organized, based on well-synthetized evidence. The authors were lucid in their style of writing, making it easy to read and understand the message, portrayed in the manuscript. Besides, the methodology design was appropriately implemented within the study. However, many of the topics are very concisely covered. This manuscript provided a comprehensive analysis of current knowledge in this field. Moreover, this research has futuristic importance and could be potential for future research. However, major concerns of this manuscript are with the introductive, methodological and discussive sections: for these reasons, I have major comments for these sections, for improvement before acceptance for publication. The article is accurate and provides relevant information on the topic and I have some major points to make, that may help to improve the quality of the current manuscript and maximize its scientific impact. I would accept this manuscript if the comments are addressed properly.

Response 4: Thank you for your kind suggestion. Your suggestion has been very helpful. We have earnestly addressed your suggestions, so if there are any further revisions needed, please do not hesitate to let me know.

As Referee 2 pointed out, we have revised the title as follows,

Micro-osteoperforations accelerate tooth movement without exacerbating the progression of root resorption in rats

P1, L2-3.

Reviewer 2 Report

Comments and Suggestions for Authors

the title should reflect the animal study, revise

very short introduction,

line 35-37, add photobiomodulation  (https://doi.org/10.1007/978-3-319-51944-9_12; Lasers Med Sci. 2023 Sep 4;38(1):200.)

figures, spell out the acronym MOPs group,

figure 4,5,6,7, add foot notes for acronyms on the graph

figure 7 need expansion of legend to explain what significant findings reader see

Comments on the Quality of English Language

needs revision

Author Response

Manuscript ID: biomolecules-2826766 - Major Revisions

Response to Reviewer 2 Comments

Thank you for giving us valuable advice and comments regarding our manuscript. We revise upon the manuscript as follows.

Reviewer 2

Comments and Suggestions for Authors

Point 1:

the title should reflect the animal study, revise

Response 1:

Thank you for the pointing out.

As you pointed out, we have revised the title as follows,

Micro-osteoperforations accelerate tooth movement without exacerbating the progression of root resorption in rats

P1, L2-3.

Point 2:

very short introduction,

Response 2:

Thank you for the pointing out.

As you pointed out, we have revised “Introduction”.

P1, L31-P2,L74.

Point 3:

line 35-37, add photobiomodulation  (https://doi.org/10.1007/978-3-319-51944-9_12; Lasers Med Sci. 2023 Sep 4;38(1):200.)

Response 3:

Thank you for the pointing out.

As you pointed out, we have added the reference in revised manuscript.

P1, L42.

Point 4:

figures, spell out the acronym MOPs group,

Response 4:

Thank you for the pointing out.

As you pointed out, we have revised figures.

Point 5:

figure 4,5,6,7, add foot notes for acronyms on the graph

Response 5:

Thank you for the pointing out.

As you pointed out, we have revised the foot note in figure 4, 5, 6, 7.

Point 6:

figure 7 need expansion of legend to explain what significant findings reader see

Response 6:

Thank you for the pointing out.

As you pointed out, we have revised figure 7.

Reviewer 3 Report

Comments and Suggestions for Authors

The manuscript is well written, and it is easy to follow. Furthermore, it depicts highly interesting insights. Nevertheless, I have identified some issues that I would like to share with the authors:

1.      In my opinion Figure 4 is incorrect as it is. Please correct me if am wrong, but the authors only collected the data form each day. Thus, the lines correspond to a artefact created by the software. If the authors wish to include the graph lines, they should be a module established and verified by the authors. Otherwise, the authors may simply display the dots of the collected data. However, this will impair the manuscript impact. Do the author understand my point of view? Furthermore, if the authors use a linear model, in my opinion a tooth movement rate should be calculated and discussed.

2.      Figure 5 A does not contain any scale.

3.      Considering that the animals are being subjected to oral intervention (during a period of 14 days), what type of food is provided? Please add details, explaining how it does not interfere with this issue. Inluding the difference to the control group. Namely, if the animals lost weight.

4.      Conclusion section is extremely reductive. I recommend the authors to provide an overview of the possible impacts of these findings in human applications, including the foreseen economical, pain, treatment duration impact.

Minor revisions:

Line 113, please separate units from numerical values, except percentage.

Line 113, percentages that describe ratios between compounds should depict (v/v) or (w/v) or (w/w).

Line 139, please revise degrees.

Author Response

Manuscript ID: biomolecules-2826766 - Major Revisions

Response to Reviewer 3 Comments

Thank you for giving us valuable advice and comments regarding our manuscript. We revise upon the manuscript as follows.

Reviewer 3

Comments and Suggestions for Authors

The manuscript is well written, and it is easy to follow. Furthermore, it depicts highly interesting insights. Nevertheless, I have identified some issues that I would like to share with the authors:

  1. In my opinion Figure 4 is incorrect as it is. Please correct me if am wrong, but the authors only collected the data form each day. Thus, the lines correspond to a artefact created by the software. If the authors wish to include the graph lines, they should be a module established and verified by the authors. Otherwise, the authors may simply display the dots of the collected data. However, this will impair the manuscript impact. Do the author understand my point of view? Furthermore, if the authors use a linear model, in my opinion a tooth movement rate should be calculated and discussed.

Response 1:

Response 1: Thank you for pointing out. We agree with this comment. Therefore, we have modified Figure 4.

  1. Figure 5 A does not contain any scale.

Response 2: Thank you for pointing this out. We agree with this comment. Therefore, we have modified Figure 5 and have added scales.

  1. Considering that the animals are being subjected to oral intervention (during a period of 14 days), what type of food is provided? Please add details, explaining how it does not interfere with this issue. Inluding the difference to the control group. Namely, if the animals lost weight.

Response 3: Thank you for pointing this out. Powdered food was provided to all the animals during the experimental period. There were no excessive weight changes in all animals. Therefore, we added a sentence “During the experimental period, powered food was provided for all the animals.” To P.2, line 84 and a sentence “There were no excessive body weight changes in all animals.” to P.5 line 193.

  1. Conclusion section is extremely reductive. I recommend the authors to provide an overview of the possible impacts of these findings in human applications, including the foreseen economical, pain, treatment duration impact.

Response 4: Thank you for pointing this out. We agree with this comment. Therefore, we added the following sentences to conclusion. “MOPs may prevent patient frustration, increased risk of dental caries, periodontitis, and the progression of root resorption by extended orthodontic treatment through accelerating orthodontic tooth movement and shortening treatment period.” 

P.12, L.410-412.

Minor revisions:

Line 113, please separate units from numerical values, except percentage.

Line 113, please separate units from numerical values, except percentage.

Thank you for pointing this out. We revised.

P.4, L.127.

Line 113, percentages that describe ratios between compounds should depict (v/v) or (w/v) or (w/w).

Line 113, percentages that describe ratios between compounds should depict (v/v) or (w/v) or (w/w).

Thank you for pointing this out. We added “(w/v)”.

P4. L.127.

Line 139, please revise degrees.

Response:

Line 139, please revise degrees.

Thank you for pointing this out. We revised.

P4, L.153.

As Referee 2 pointed out, we have revised the title as follows,

Micro-osteoperforations accelerate tooth movement without exacerbating the progression of root resorption in rats

P1, L2-3.

Round 2

Reviewer 1 Report

Comments and Suggestions for Authors

The authors have significantly improved the manuscript.

Author Response

Manuscript ID: biomolecules-2826766 - Minor Revisions

Response to Reviewer 1 Comments

Thank you for giving us valuable advice and comments regarding our manuscript. We revise upon the manuscript as follows.

Reviewer 1

Comments and Suggestions for Authors

The authors have significantly improved the manuscript.

Response:

Thank you for your kind review. I appreciate it.

Reviewer 2 Report

Comments and Suggestions for Authors

Thank you for the revisions 

Comments on the Quality of English Language

Needs copy editing 

Author Response

Manuscript ID: biomolecules-2826766 - Minor Revisions

Response to Reviewer 2 Comments

Thank you for giving us valuable advice and comments regarding our manuscript. We revise upon the manuscript as follows.

Reviewer 2

Comments and Suggestions for Authors

Thank you for the revisions 

Response :

Thank you for your kind review. I appreciate it.

Reviewer 3 Report

Comments and Suggestions for Authors

I acknowledge the author improvements. I still think that the conclusion section is reductive however it does not impede the publication of this manuscript.

Author Response

Manuscript ID: biomolecules-2826766 - Minor Revisions

Response to Reviewer 3 Comments

Thank you for giving us valuable advice and comments regarding our manuscript. We revise upon the manuscript as follows.

Reviewer 3

Comments and Suggestions for Authors

I acknowledge the author improvements. I still think that the conclusion section is reductive however it does not impede the publication of this manuscript.

Response : Thank you for pointing this out. We agree with this comment. Therefore, we added the following sentences to conclusion as follows;.

From this study, it was found that MOPs facilitate OTM during orthodontic treatment without exacerbating root resorption. This result demonstrates that MOPs, which are the least invasive and painful among surgical approaches such as corticotomy, do not exacerbate root resorption, an unintended consequence of orthodontic treatment, and shorten the treatment duration. Consequently, this is good news for adults who tend to experience slow tooth movement and prolonged treatment duration, and an increase in the number of adults undergoing orthodontic treatment is expected.

P.12, L.409-419.
